# FLIS: Clustered Federated Learning via Inference Similarity for Non-IID Data Distribution

## Abstract

Classical federated learning approaches yield significant performance degradation in the presence of Non-IID data distributions of participants. When the distribution of each local dataset is highly different from the global one, the local objective of each client will be inconsistent with the global optima which incur a drift in the local updates. This phenomenon highly impacts the performance of clients. This is while the primary incentive for clients to participate in federated learning is to obtain better personalized models. To address the above-mentioned issue, we present a new algorithm, FLIS, which groups the clients population in clusters with jointly trainable data distributions by leveraging the inference similarity of clients' models. This framework captures settings where different groups of users have their own objectives (learning tasks) but by aggregating their data with others in the same cluster (same learning task) to perform more efficient and personalized federated learning. We present experimental results to demonstrate the benefits of FLIS over the state-of-the-art benchmarks on CIFAR-100/10, SVHN, and FMNIST datasets.

## 1 Introduction

Federated learning (FL) is a recently proposed distributed training framework that enables distributed users to collaboratively train a shared model under orchestration of a central server without compromising the data privacy of users [1]. While brings us great potential, FL faces challenges in practical settings. For example, due to the statistical heterogeneity (Non-IIDness) of the distribution of the distributed data, learning a single deep learning model on the server as in [2, 3, 4] lacks flexibility and personalization and yield poor performance [5, 6, 7]. Due to the Non-IIDness, it turns out that some of participants gain no benefit by participating in FL since the global shared model is less accurate than the local models that they can train on their own [8, 9]. This is while one of the main incentives for clients to participate in FL is to improve their personal model performance. Specially, for those clients who have enough private data, there is not much benefit to participate in FL [7]. Personalized FL under data heterogeneity was also realized via performing clustering [10, 11]. Clustered-FL addresses this problem by grouping clients into separate clusters based on either geometric properties of the FL loss surface [11] or based on weights of models or model update comparisons at the server side [12].

Motivated by the above-mentioned, it is therefore, natural to ask the question: *How can one benefit the most from FL when each participant has a varying amount of data coming from distinct distributions that is a black box to others?* This is the canonical question that we will answer in this paper. In the current paper, we propose a clustered federated learning algorithm where the clients are partitioned into different clusters depending upon their data distributions. Our goal is to group the clients with similar data distributions in the same cluster without having access to their private data and then train models for every cluster of users. The main idea of our algorithm is a strategy that alternates between estimating the cluster identities and maximizing the inference similarity at the server side. Our main contributions can be summarized as follows.

- We propose the idea of inference similarity as a way for the central server to identify cluster ID of clients that have similar data distributions without requiring any access to the private data of clients. This way, clients in the same cluster can benefit from each other's training without the corruptive influence of clients with unrelated data distributions.

- Our algorithm can constitute joint and disjoint clusters and does not require the number of clusters to be known apriori. Further, it is effective both in Non-IID and IID regimes. In contrast, prior clustered FL works [10, 11] considers a pre-defined number of clusters (models) on the server and assign a hard membership ID to the clients. In such settings, the proposed method could perform poorly for many of the clients under pathological highly skewed Non-IID data which requires more number of clusters, and slightly skewed Non-IID data which requires fewer clusters since we cannot know how many unique data distributions the client's datasets are drawn from.

- We perform extensive experimental studies to evaluate FLIS and verify its performance for Non-IID FL. In particular, we demonstrate that the proposed approach can significantly outperform the existing state-of-the-art (SOTA) global model FL benchmarks by up to $\sim 40\%$, and the SOTA personalized FL baselines by up to $\sim 30\%$.

## 2 Federated Learning with Clustering

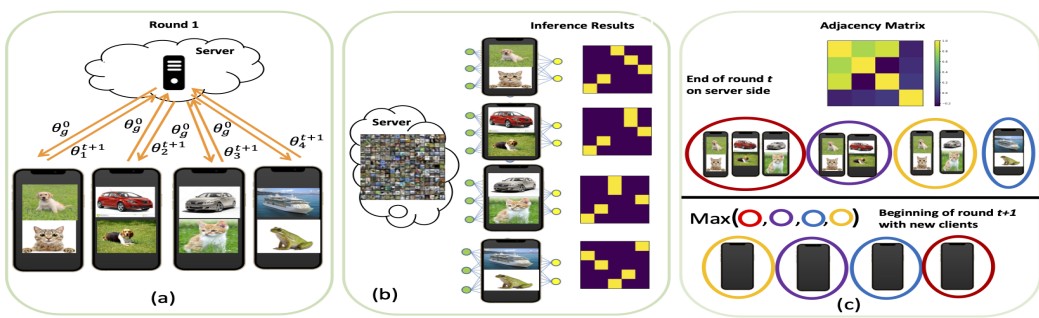

**Figure 1:** A toy example showing the overview of FLIS algorithm. (a) The server sends the initial global model to the clients at round 1. The clients update the received model using their local data and send back their updated models to the server. (b) The server captures the inference results on its own small dataset. Then according to the similarity of the inference results, the clients are clustered. In this example, clients 1 and 2, and 3 are yielding more similar inference results compared to client 4. (c) The server uses inference similarity results to constitute the adjacency matrix and identify their cluster IDs via hard thresholding or hierarchical clustering and does model averaging within each cluster.

---

**Algorithm 1:** The FLIS (DC) framework

---

**Require:** Number of available clients $N$, sampling rate $R \in (0, 1]$, Data on the server $D^{Server}$, clustering threshold $\beta$

**Init:** Initialize the server model with $\theta_g^0$

1 **Def** `FLIS_DT`:
2    **for** each round $t = 0, 1, 2, \ldots$ **do**
3      $n \leftarrow \max(R \times N, 1)$
4      $\mathcal{S}_t \leftarrow \{k_1, \ldots, k_n\}$ random set of $n$ clients
5      **for** each client $k \in \mathcal{S}_t$ **in parallel do**
6        **if** $t = 0$ **then**
7          download $\theta_g^0$ from the server and start training, i.e. $\theta_{k,j_t^*}^t = \theta_g^0$
8        **else**
9          download clusters $\theta_{g,j_t}^t, j_t = 1, \ldots, T_t$ from the server and select the best cluster according to $\theta_{k,j_t^*}^t = \arg\min L_k(D_k^{test}; \theta_{g,j_t}^t)$
10        $\theta_{k,j_t^*}^{t+1} \leftarrow \text{ClientUpdate}(C_k; \theta_{k,j_t^*}^t)$                  `// SGD training`
11      $\{C_{j_{t+1}}\}_{j_{t+1}=1}^{T_{t+1}} = \text{ISC}(D^{Server}, \{\theta_{k,j_t^*}^{t+1}\}_{k=1,\ldots,n})$    `// dynamically clustering clients via inference similarity`
12      $\theta_{g,j_{t+1}}^{t+1} = \sum_{k \in C_{j_{t+1}}} |D_k| \theta_{k,j_t^*}^{t+1} / \sum_{k \in C_{j_{t+1}}} |D_k|$

---

## 2.1 Overview of FLIS Algorithm

In this section, we provide details of our algorithm. We name this algorithm Federated Learning by Inference Similarity (FLIS). FLIS is able to form both joint dynamic clusters with soft membership ID, named as FLIS (DC) and disjoint hierarchically formed clusters with hard membership ID, named as FLIS (HC). The overview of FLIS (DC) which forms joint clusters is sketched in Figure 1 and presented in Algorithm 1, and 2. The overview of FLIS (HC) which forms disjoint clusters is presented in Algorithm 3. The first round of the algorithm starts with a random initial model parameters $\theta_g$. In the t-th iteration of FLIS, the central server samples a random subset of clients $\mathcal{S}_t \subseteq [N]$ ($N$ is the total number of clients), and broadcasts the current model parameters $\{\theta_{g,j_i}^t\}_{i=1}^T$ to the clients in $\mathcal{S}_t$. We recall that the local objective $L_k$ is typically defined by the empirical loss over local data. Each client then estimates its cluster identity via finding the model parameter that yields minimum loss on its test data, i.e., $\theta_{k,j_t^*}^t = \mathrm{argmin}_j \, L_k(D_k^{server}; \theta_{g,j_t}^t)$. Then the clients perform $\mathcal{T}$ steps of stochastic gradient descent (SGD) updates, get the updated model, and send their model parameters, $\{\theta_k^{t+1}\}_{k=1}^{\|\mathcal{S}_t\|}$, to the server. After receiving the model parameters from all the participating clients, the server then leverages inference similarity as a way to form dynamic clusters of clients that have similar data distributions. Finally, the server collects all the parameters from clients who are in the same cluster and averages the model parameters of each cluster.

---

**Algorithm 2:** Inference Similarity Clustering (ISC)

---

**Require:** Data on the server $D^{Server}$, $\beta$
**Return:** The formed clusters $\{C_j\}$

1 **Function** ISC($D^{Server}$, $\{\theta_{k,j_t^*}^{t+1}\}_{k=1,\ldots,n}$)**:**

2     $B_k = F_k(D^{Server}; \theta_{k,j_t^*}^{t+1})$     // $F_k$ is the function defined over the client model

3     $A_{i,j} = \frac{\|B_i \odot B_j\|_F}{\|B_i\|_F \|B_j\|_F}; i, j = 1, \ldots, n$     // Server constructs the adjacency matrix

4     $\tilde{A}_{i,j} = \Gamma(A_{i,j}) = \mathrm{Sign}(A_{i,j} - \beta)$ // Server applies hard thresholding and does joint clustering

5     **Return** $\{C_{j_{t+1}}\}_{j_{t+1}=1}^{T_{t+1}}$

---

**Algorithm 3:** The FLIS (HC) framework

---

**Require:** Number of available clients $N$, sampling rate $R \in (0, 1]$, Data on the server $D^{Server}$, clustering threshold $\beta$
**Init:** Initialize the server model with $\theta_g^0$

1 **Def** FLIS_HC**:**

2     **for** each round $t = 0, 1, 2, \ldots$ **do**

3         **if** $t = 1$ **then**

4             All clients receive the initial server model $\theta_g^0$, perform local update and send back the updated models to the server.

5             $\mathbf{A} \leftarrow$ server forms $\mathbf{A}$ based on $A_{i,j}$ defined in Subsection B.

6             $\{C_1, \ldots, C_j\} = \mathrm{HC}(\mathbf{A}, \beta)$     // performing hierarchical clustering to obtain the clusters

7             $\theta_{g,j}^0 \leftarrow \theta_g^0$     // initializing all clusters with $\theta_g^0$

8         **else**

9             $n \leftarrow \max(R \times N, 1)$

10            $\mathcal{S}_t \leftarrow \{k_1, \ldots, k_n\}$ random set of $n$ clients

11         **for** each client $k \in \mathcal{S}_t$ **in parallel do**

12             Each client $k$ receives its cluster model from the server $\theta_{g,j_k}^t$, $j = 1, \ldots, T$

13             $\theta_{k,j_k}^{t+1} \leftarrow \mathrm{ClientUpdate}(C_k; \theta_{k,j_k}^t)$     // SGD training

14         $\theta_{g,j}^{t+1} = \sum_{k \in C_j} |D_k| \theta_{k,j_k}^{t+1} / \sum_{k \in C_j} |D_k|$

---

## 2.2 Clustering Clients

Herein, we are aiming to find the clients with similar data distributions without requiring any prior knowledge about the data distributions. In doing so, we assume that the server has some real or synthetic data on its own [1]. The server then performs inference on each client model and obtain

---

[1] The number of auxiliary samples used for forming the clusters at the server is 2500.

**Table 1:** Test accuracy comparison across different datasets for Non-IID label skew $(20\%)$, and $(30\%)$.

| Algorithm | FMNIST | CIFAR-10 | CIFAR-100 | SVHN |
|---|---|---|---|---|
| | | Non-IID label skew (20%) | | |
| SOLO | $95.92 \pm 0.57$ | $79.22 \pm 1.67$ | $32.28 \pm 0.23$ | $79.72 \pm 1.37$ |
| FedAvg | $77.3 \pm 4.9$ | $49.8 \pm 3.3$ | $53.73 \pm 0.50$ | $80.2 \pm 0.8$ |
| FedProx | $74.9 \pm 2.6$ | $50.7 \pm 1.7$ | $54.35 \pm 0.84$ | $79.3 \pm 0.9$ |
| FedNova | $70.4 \pm 5.1$ | $46.5 \pm 3.5$ | $53.61 \pm 0.42$ | $75.4 \pm 4.8$ |
| Scafold | $42.8 \pm 28.7$ | $49.1 \pm 1.7$ | $54.15 \pm 0.42$ | $62.7 \pm 11.6$ |
| LG | $96.80 \pm 0.51$ | $86.31 \pm 0.82$ | $45.98 \pm 0.34$ | $92.61 \pm 0.45$ |
| PerFedAvg | $95.95 \pm 1.15$ | $85.46 \pm 0.56$ | $60.19 \pm 0.15$ | $93.32 \pm 2.05$ |
| IFCA | $97.15 \pm 0.01$ | $87.99 \pm 0.15$ | $71.84 \pm 0.23$ | $95.42 \pm 0.06$ |
| CFL | $77.93 \pm 2.19$ | $51.11 \pm 1.01$ | $40.29 \pm 2.23$ | $73.62 \pm 1.76$ |
| **FLIS (DC)** | $\mathbf{97.64 \pm 0.38}$ | $\mathbf{89.47 \pm 0.92}$ | $\mathbf{73.91 \pm 0.29}$ | $\mathbf{95.65 \pm 0.17}$ |
| **FLIS (HC)** | $\mathbf{97.45 \pm 0.08}$ | $\mathbf{89.35 \pm 0.46}$ | $\mathbf{73.20 \pm 0.31}$ | $\mathbf{95.48 \pm 0.21}$ |
| | | Non-IID label skew (30%) | | |
| SOLO | $93.93 \pm 0.10$ | $65 \pm 0.65$ | $22.95 \pm 0.81$ | $68.70 \pm 3.13$ |
| FedAvg | $80.7 \pm 1.9$ | $58.3 \pm 1.2$ | $54.73 \pm 0.41$ | $82.0 \pm 0.7$ |
| FedProx | $82.5 \pm 1.9$ | $57.1 \pm 1.2$ | $53.31 \pm 0.48$ | $82.1 \pm 1.0$ |
| FedNova | $78.9 \pm 3.0$ | $54.4 \pm 1.1$ | $54.62 \pm 0.91$ | $80.5 \pm 1.2$ |
| Scafold | $77.7 \pm 3.8$ | $57.8 \pm 1.4$ | $54.90 \pm 0.42$ | $77.2 \pm 2.0$ |
| LG | $94.21 \pm 0.40$ | $76.58 \pm 0.16$ | $35.91 \pm 0.20$ | $87.69 \pm 0.77$ |
| PerFedAvg | $92.87 \pm 2.67$ | $77.67 \pm 0.19$ | $56.42 \pm 0.41$ | $91.25 \pm 1.47$ |
| IFCA | $95.22 \pm 0.03$ | $80.95 \pm 0.29$ | $67.39 \pm 0.27$ | $93.02 \pm 0.15$ |
| CFL | $78.44 \pm 0.23$ | $52.57 \pm 3.09$ | $35.23 \pm 2.72$ | $73.97 \pm 4.77$ |
| **FLIS (DC)** | $\mathbf{95.95 \pm 0.51}$ | $\mathbf{82.25 \pm 1.12}$ | $\mathbf{68.36 \pm 0.12}$ | $\mathbf{93.08 \pm 0.22}$ |
| **FLIS (HC)** | $\mathbf{95.35 \pm 0.16}$ | $\mathbf{82.17 \pm 0.22}$ | $\mathbf{67.51 \pm 0.23}$ | $\mathbf{93.10 \pm 0.20}$ |

a $\tilde{M} \times \tilde{N}$ matrix, $B_k = F_k(D^{server}; \theta_{k,j_t^*}^t)$, $k = 1, ..., \|\mathcal{S}_t\|$, where $\tilde{N}$, and $\tilde{M}$ are the number of final neurons of the last fully connected layer (classification layer), and the number of data on the server, respectively. Note that, the columns of $B_k$ can be one-hot or soft labels. Using $B_k$, the server constructs an adjacency matrix as $A_{i,j} = \frac{\|B_i \odot B_j\|_F}{\|B_i\|_F \|B_j\|_F}$, where $i, j = 1, ..., \|\mathcal{S}_t\|$, and $\odot$ stands for Hadamard product. Having the adjacency matrix $A_{i,j}$, as mentioned earlier, depending on whether forming joint clusters are of interest or the disjoint ones, we propose two different clustering approaches. For FLIS (DC) that constructing joint clusters on the server is of interest, we define a hard thresholding operator $\Gamma$ which is applied on $A_{i,j}$ and yields $\tilde{A}_{i,j} = \Gamma(A_{i,j}) = \text{Sign}(A_{i,j} - \beta)$, with $\beta$ being a threshold value. Now, making use of $\tilde{A}_{i,j}$, the server can form joint clusters of interest by putting indices of the positive entries in each row of $\tilde{A}_{i,j}$ in the same cluster as is shown in the toy example in Fig 1. In FLIS (DC), in each round 10 clusters is formed which is equal to the number of participant clients in each round. For FLIS (HC), having $\tilde{A}_{i,j}$ in hand, the server can group the clients by employing hierarchical clustering (HC) [13] as presented in Algorithm 3 ). It is noteworthy that in FLIS (HC) the number of formed clusters are fixed and depends upon the distance threshold of HC which is a hyperparameter.

# 3 Experiments

## 3.1 Experimental Settings

**Datasets and Models.** We conduct experiments on CIFAR-10, CIFAR-100, SVHN, and Fashion MNIST (FMNIST) datasets. For each dataset we considered three different federated heterogeneity settings as in [14]: Non-IID label skew $(20\%)$, Non-IID label skew $(30\%)$, and Non-IID Dir$(0.1)$. We used Lenet-5 architecture for CIFAR-10, SVHN, and FMNIST datasets, and ResNet-9 architecture for CIFAR-100 dataset.

**Baselines.** To show the effectiveness of the proposed method, we compare the results of our algorithm against SOTA personalized FL methods i.e., LG-FedAvg [15], Per-FedAvg [5], IFCA [10], CFL [11], as well as methods targeting to learn a single global model i.e., FedAvg [2], FedProx [16], FedNova [4], and SCAFFOLD [3]. We also compare our results with another baseline named SOLO, where each client trains a model on its own local data without taking part in FL. Our code is available at https://github.com/anonresearcher1/alg-novel-flis.

**Performance Comparison.** Table 1, and 2, show the average final top-1 test accuracy of all clients for all the SOTA algorithms under Non-IID label skew $(20\%)$, Non-IID label $(30\%)$, and Non-IID Dir$(0.1)$ setups, respectively. In these tables we report the results of the two proposed

**Table 2:** Test accuracy comparison for Non-IID Dir(0.1).

| Algorithm | FMNIST | CIFAR-10 | CIFAR-100 |
|-----------|--------|----------|-----------|
| SOLO | $69.71 \pm 0.99$ | $41.68 \pm 2.84$ | $16.83 \pm 0.51$ |
| FedAvg | $82.91 \pm 0.83$ | $38.22 \pm 3.28$ | $44.52 \pm 0.42$ |
| FedProx | $84.04 \pm 0.53$ | $42.29 \pm 0.95$ | $45.52 \pm 0.72$ |
| FedNova | $84.50 \pm 0.66$ | $40.25 \pm 1.46$ | $46.52 \pm 1.34$ |
| Scafold | $10.0 \pm 0.0$ | $10.0 \pm 0.0$ | $43.73 \pm 0.89$ |
| LG | $74.96 \pm 1.41$ | $49.65 \pm 0.37$ | $23.59 \pm 0.26$ |
| PerFedAvg | $80.29 \pm 2.00$ | $53.58 \pm 1.57$ | $33.94 \pm 0.41$ |
| IFCA | $85.01 \pm 0.30$ | $51.16 \pm 0.49$ | $47.67 \pm 0.28$ |
| CFL | $74.13 \pm 0.94$ | $42.30 \pm 0.25$ | $31.42 \pm 1.50$ |
| **FLIS (DC)** | $\mathbf{86.5 \pm 0.76}$ | $\mathbf{60.33 \pm 2.30}$ | $\mathbf{53.85 \pm 0.56}$ |
| **FLIS (HC)** | $\mathbf{85.21 \pm 0.18}$ | $\mathbf{51.18 \pm 0.21}$ | $\mathbf{49.10 \pm 0.19}$ |

clustering approaches i.e., FLIS (DC) (presented in Algorithm 1) as well as FLIS (HC) (presented in Algorithm 3). Under Non-IID settings, SOLO with zero communications cost demonstrates much better accuracy than all the global FL baselines including FedAvg, Fedprox, FedNova, and SCAFFOLD. On the other hand, each client itself may not have enough data and thus we need to better exploit the similarity among the users by clustering. This further explains the benefits of personalization and clustering in Non-IID settings. Comparing different FL approaches, we can see that FLIS (DC) consistently yields the best accuracy results among all tasks. It can outperform FedAvg by up to $\sim 40\%$.

It is apparent from table 2 for Non-IID Dir(0.1) that LG-FedAvg and Per-FedAvg perform even worse than FedAvg. The performance of CFL benchmark is close to that of FedAvg in most cases, and even worse. IFCA (with two clusters, C=2) obtained the closest results to FLIS , but FLIS consistently beats IFCA especially in Non-IID Dir(0.1) by a large margin. *FLIS shows superior learning performance over the SOTA on more challenging tasks.* For instance, FLIS, is noticeably better than IFCA for CIFAR-10 which is a harder task compared to FMNIST and SVHN by up to $\sim 10\%$ in Non-IID Dir(0.1). As a final note, we also studied the impact of constructing disjoint clusters. HC by extracting disjoint clusters, seems to be slightly deteriorating the performance of FLIS, even though it still remains to be on par with the best performing baselines.

### 3.2 Communication Efficiency

### 3.2.1 What is the Required Communication Cost/Round to Reach a Target Test Accuracy?

We additionally compare the SOTA baselines in terms of the number of communication round/Communication cost that is required to reach a specific target accuracy. Table 3 reports the required number of communication round and communication cost to reach the designated target test accuracies for Non-IID label skew (20%) and Non-IID label skew (30%), respectively. As is observed from the table, in all scenarios, FLIS has the minimum communication round. For instance, 37 number of rounds are sufficient for FLIS to achieve the target accuracy of 50% for Non-IID label skew (20%) in CIFAR-100, whereas some other baselines, e.g. Per-FedAvg requires $\sim 4\times$ more communication rounds and global model FL baselines are the most expensive ones in general. IFCA requires the closest number of rounds compared to FLIS to reach the target test accuracies in general. We attribute this to the fact that by grouping the clients with similar data distributions in the same clusters, the setting tends to mimic the IID setting, which means faster convergence in fewer communication round. Note that " $--$ " means the baseline was not able to reach the target accuracy. This characteristics of FLIS (HC) is desirable in practice as it helps to reduce the communication overhead in FL systems in two ways: first, it converges fast and second, rather than communicating all clusters (models) with the clients, the server will receive the cluster ID from each client and then only send the corresponding cluster to each client.

### 3.3 Impact of Hyper-parameter Changes

Herein, we study the impact of a few important hyper-parameters on the performance of FLIS as in the following.

**The influence of the inference similarity threshold $\beta$.** We investigate the effect of the inference similarity threshold $\beta$ on the final test accuracy. Fig. 2 visualizes the accuracy performance behavior of FLIS under different values of $\beta$, as well as the local epochs for several datasets for Non-IID (20%). We vary $\beta$ from 0 to 1. The parameter $\beta$ controls the similarity of the data distribution of clients within a cluster. Therefore, $\beta$ achieves a trade-off between a purely local and global model

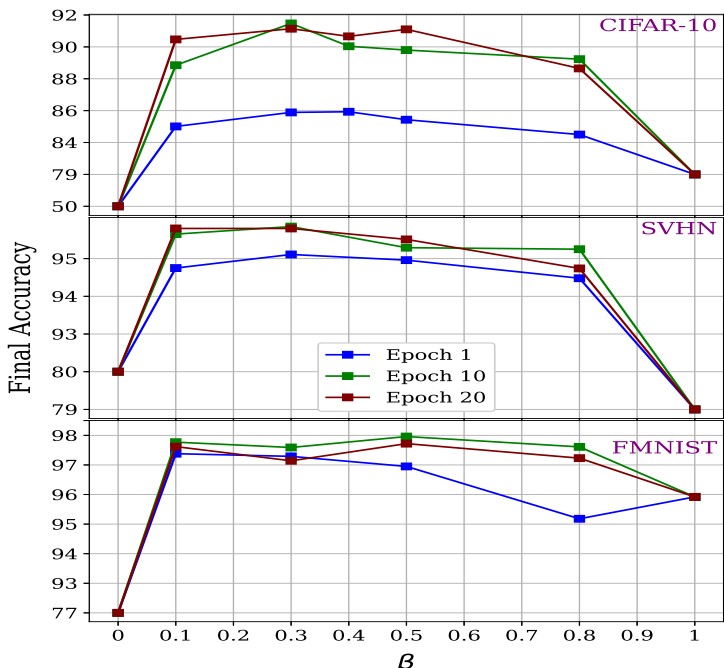

**Figure 2:** Evaluating FLIS (DC)'s accuracy performance versus the inference similarity threshold $\beta$, and number of local epoch for Non-IID label skew (20%) on CIFAR-10, FMNIST, and SVHN datasets. FLIS (DC) benefits from larger numbers of local training epochs.

**Table 3:** Comparing different FL approaches for Non-IID (20%) in terms of the required number of communication rounds, and for Non-IID (30%) in terms of the required communication cost in **Mb** to reach target top-1 average local test accuracy: communication round/communication cost.

| Algorithm | FMNIST | CIFAR-10 | CIFAR-100 | SVHN |
|---|---|---|---|---|
| Target | 80% | 70% | 50% | 75% |
| FedAvg | 200/79.36 | −−/−− | 130/4237.37 | 150/71.43 |
| FedProx | 200/71.43 | −−/−− | 115/4237.37 | 200/71.43 |
| FedNova | −−/−− | −−/−− | 120/3601.98 | 150/79.36 |
| Scafold | −−/−− | −−/−− | 82/3305.11 | −−/−− |
| LG | 13/**1.26** | 33/**2.11** | −−/−− | 16/**1.76** |
| PerFedAvg | 19/7.54 | 60/23.81 | 110/6356.06 | 39/18.65 |
| IFCA | 14/11.30 | 25/16.66 | 40/3495.19 | 17/10.71 |
| CFL | −−/−− | −−/−− | −−/−− | −−/−− |
| **FLIS (HC)** | **12**/7.53 | **24**/10.31 | **37**/1991.60 | **15**/8.73 |

and provides a trade-off between generalization and distribution heterogeneity. To delineate, when $\beta = 0$, FLIS groups all the clients into 1 cluster and the scenario reduces to FedAvg baseline. This is the reason for the significant accuracy drop at $\beta = 0$ as it is also evident from figure 2, by increasing $\beta$, FLIS becomes more strict in grouping the clients. It means FLIS only groups the clients with more amount of label/feature overlap into a cluster leading to a more personalized FL. The optimal performance for CIFAR-10, SVHN, and FMNIST are achieved at $\beta = 0.3$, $\beta = 0.3$, and $\beta = 0.5$, respectively. Finally, when $\beta$ is 1, the scenario almost reduces to SOLO baseline where each client receives the model from the server and lonely trains it on it own local data. It is noteworthy that Non-IID (30%) has the same behavior, which was not depicted here due to space limitations.

**Benefit of more local updates.** The benefits of FLIS can be further pronounced by increasing the number of local epochs. The results are shown in Figure 2. As can be seen, when the number of local epoch is 1, the clients' local updates are very small. Therefore, the training will be slow and the accuracy becomes lower compared to the bigger number of local epochs given a fixed number of communication rounds. Also, when the clients have not been trained enough, their inference results at server side would be erroneous which further causes less accurate clustering. Figure. 2, shows the performance of FLIS is coupled with local training epochs specially on more challenging tasks. In contrast, it was shown in [14] when the number of local epochs is too large, the accuracy of all non-personalized models drop which is due to severe-side averaged models drift form the clients' local models [4].

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
