# OpenReview forum: "FLIS: Clustered Federated Learning via Inference Similarity for Non-IID Data Distribution"
_NeurIPS.cc/2022/Workshop/Federated_Learning — FL-NeurIPS 2022 Poster_

### Official Review · Reviewer_3PWe · 2022-10-16

In this work, the authors propose a new method of clustered federated learning. Based on the inference similarity on a public dataset, the method groups local models into clusters and aggregates each cluster. Results show that the proposed method has advantages over several global federated learning methods and previous cluster-based federated learning methods.

My main concern is that the improvement compared to IFCA is quite small, and the proposed method requires the availability of a public dataset on the server side while IFCA does not. Meanwhile, the experiments lack many personalized FL baselines in recent years, such as pFedMe, FedFOMO, FedRep, FedMD, etc.

Typo: In all the tables, "Scafold" should be "SCAFFOLD".

---

### Official Review · Reviewer_MPfF · 2022-10-16
**Promising algorithms, but the writing lacks clarity and a lot of details are missing**

The authors propose an inference based clustering framework for improving FL. Instead of clustering on the models, the proposed method performs clustering on the inferences on a synthetic dataset at the server. The authors propose two algorithms based on fixed clustering and dynamic clustering and show empirical results on some image datasets. The algorithms look promising, but the paper clearly needs a lot of re-writing with a lot of details not included, or missing (described below).

## Questions:
- What is soft membership and hard membership? As far as I understand, hard membership implies that the cluster assignments are fixed (at the beginning for HC) and for soft membership, it may vary at each epoch. Is this correct? It might be useful to give more details on such terminology.
- The difference between DC and HC as far as I see, is that, in HC, the clusters are decided at the beginning of the training and are kept fixed. Whereas for DC, it changes at every round. What is the rationale behind HC? Clearly, the initial clustering is only optimal for $w_0$, and the clustering may be very different for $w^*$ (which is the true global optimum). Is it because of the cost of clustering at every epoch? The authors should clearly state the differences and advantage of one over the other. The algorithm descriptions and the writing can be improved along these lines.
- In the HC algorithm, line 5, the authors refer to subsection B for details about computing $A$. Is it the same as in algorithm 2? There is no subsection B in the submission. Also, in line 3, should it be t=0, instead of t=1? In the line above, the training starts at t=0.
- What is the purpose of doing hierarchical clustering, and why not any other clustering? This is not discussed clearly.
- In line 88, authors mention that in DC, 10 clusters are formed at each round. Why is this the case? As far as I understand, the authors do not want to assume the number of clusters at all (as compared to earlier work).
- The authors mention very little about the auxiliary dataset/synthetic dataset assumption at the server. This is a key ingredient for the proposed algorithm. What distribution does this dataset come from? If the workers data is split in some non-iid way, is the synthetic dataset from the original distribution, or some mixture of the workers distributions, etc? What is the practicality of assuming such a synthetic dataset? What if one can use just random gaussian noise as the synthetic dataset? Does the algorithm still work?
- The authors only consider model homogeneous situation in both HC and DC. The clustering on the inferences is a very nice idea which can easily be extended to the model heterogeneous situation where each worker can have a different model architecture. The proposed algorithms in their current form are not applicable to the heterogeneous case. It would be interesting to see if the authors discuss this and how it can be extended.

### Experimental setup questions:
- The experimental setup doesn't clearly describe the details of the data split. The authors should describe it in the supplementary, or include references of earlier works for the label-skew and Dir setups.
- Also, the size of the synthetic dataset is fixed at 2500. How does this size compare to that of the size at each worker? What would happen if the synthetic dataset size is small? More ablations on this would be helpful.
- Moreover, the number of workers in each experiment is not mentioned.
- The authors mention that SOLO performs better than FedAvg FedProx, etc from Table 1 and 2. However, it only holds for FEMNIST and CIFAR10 in Table 1 (under label skew). This conclusion doesn't hold for CIFAR100 and SVHN in Table 1, and for all datasets in Table 2.
- What is the fraction of clients used in every communication round?

---

### Decision · Program_Chairs · 2022-10-20

Accept (Poster)